# Driving Scene Understanding with Traffic Scene-Assisted Topology Graph Transformer

## ABSTRACT

Driving scene topology reasoning aims to understand the objects present in the current road scene and model their topology relationships to provide guidance information for downstream tasks. Previous approaches fail to adequately facilitate interactions among traffic objects and neglect to incorporate scene information into topology reasoning, thus limiting the comprehensive exploration of potential correlations among objects and diminishing the practical significance of the reasoning results. Besides, the lack of constraints on lane direction may introduce erroneous guidance information and lead to a decrease in topology prediction accuracy. In this paper, we propose a novel topology reasoning framework, dubbed TSTGT, to address these issues. Specifically, we design a divide-and-conquer topology graph Transformer to respectively infer the lane-lane and lane-traffic topology relationships, which can effectively aggregate the local and global object information in the driving scene and facilitate the topology relationship learning. Additionally, a traffic scene-assisted reasoning module is devised and combined with the topology graph Transformer to enhance the practical significance of lane-traffic topology. In terms of lane detection, we develop a point-wise matching strategy to infer lane centerlines with correct directions, thereby improving the topology reasoning accuracy. Extensive experimental results on Openlane-V2 benchmark validate the superiority of our TSTGT over state-of-the-art methods and the effectiveness of our proposed modules.

## CCS CONCEPTS

• **Computing methodologies** → **Autonomous Driving**.

## KEYWORDS

Autonomous Driving, Lane Perception, Topology reasoning

**ACM Reference Format:**
Anonymous Submission and Paper ID: 4230. 2024. Driving Scene Understanding with Traffic Scene-Assisted Topology Graph Transformer. In *Proceedings of Make sure to enter the correct conference title from your rights confirmation emai (Conference acronym 'XX).* ACM, New York, NY, USA, 10 pages. https://doi.org/XXXXXXX.XXXXXXX

## 1 INTRODUCTION

Driving scene topology reasoning aims to detect lane and traffic elements in multi-view images captured by onboard cameras, and to construct the topology relationships between them. This emerging task of scene understanding provides a more natural integration for perception and planning tasks in autonomous driving, attracting significant attention in the research community. The driving scene topology reasoning task can provide information about drivable areas and traffic signals, so as to offer clear navigation signals for downstream tasks such as motion prediction and planning. Compared to the lane detection task and the 3D object detection task, driving scene topology reasoning is more challenging because of the difficulty in understanding the topology relationships of objects in complex scenes.

Recently, Wang et al. [44] proposed a dataset called Openlane-V2 that defines the objectives of the driving scene topology reasoning task. Specifically, given multi-view images, topology reasoning aims to learn the vectorized road graph between centerlines and traffic elements. It includes four sub-tasks, namely lane centerline detection, traffic element detection, lane-lane topology reasoning and lane-traffic topology reasoning. Some explorations are advancing the field. For instance, Li et al. [26] proposed the TopoNet, which uses a GCN [23] to construct topology relationships between heterogeneous features. Wu et al. [48] proposed the state-of-the-art TopoMLP, where the concept of "first-detect-then-reason" is introduced and a simple MLP is employed to build the topology relationships of target objects.

Despite the impressive performance of some existing methods, there are still some drawbacks that need to be addressed. **First**, an important challenge in this task is how to construct the topology relationships of lane-lane and lane-traffic, while previous methods are not perfect in this aspect. As shown in Fig.1 (a), TopoNet [26] adopted a graph convolutional network (GCN) to construct topology relationships. However, using a single graph model to simultaneously build topology relationships between homogeneous and heterogeneous features may lead to confusion of object information. In addition, simultaneous detection and reasoning may result in the loss of object information. Therefore, TopoMLP [48] proposed the strategy of "first-detect-then-reason" and utilized two separate sets of MLPs to predict the topology relationships between different types of objects, as shown in Fig.1 (b). Nevertheless, from a graph perspective, MLPs can only aggregate information between pairs of nodes and cannot capture information from local and global nodes. This makes it difficult to effectively acquire intrinsic connections between objects. **Second**, solely relying on data without considering the practical significance of objects to construct topology relationships cannot effectively address complex driving scenes. Effective constraints on topology reasoning results based on traffic rules and the potential topology relationships provided by the

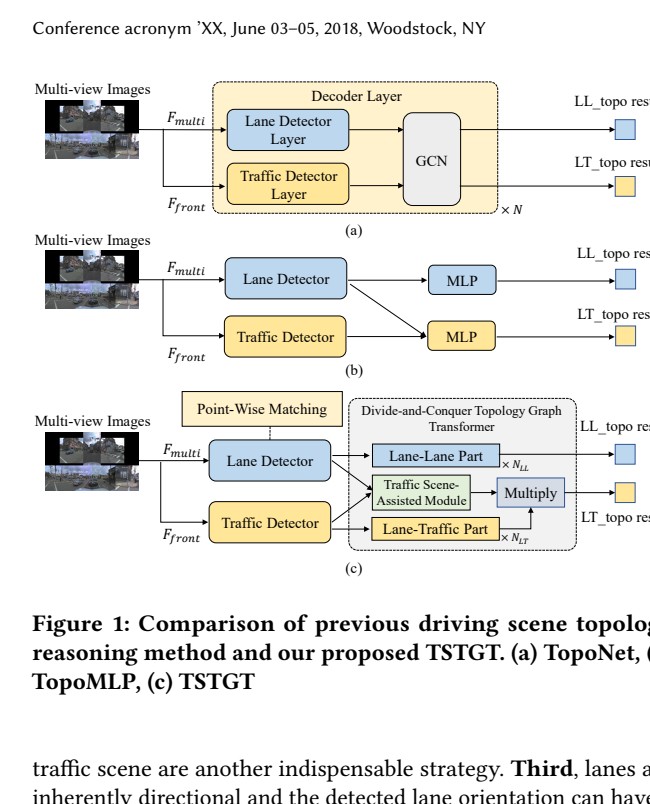

**Figure 1: Comparison of previous driving scene topology reasoning method and our proposed TSTGT. (a) TopoNet, (b) TopoMLP, (c) TSTGT**

traffic scene are another indispensable strategy. **Third**, lanes are inherently directional and the detected lane orientation can have a significant impact on the results of lane-lane topology reasoning. Previous methods, such as MapTR [30], employed hierarchical bipartite matching to constrain the orientation of HD Map objects. In contrast, designing an approach to ensure the correct lane orientation in the driving scene topology reasoning task remains to be developed.

In this study, we propose TSTGT (**T**raffic **S**cene-Assisted **T**opology **G**raph **T**ransformer), an innovative driving scene topology reasoning network, to address the aforementioned issues, as illustrated in Fig.1 (c). Inspired by GPS [40], we first utilize lane embeddings and traffic element embeddings as graph node features inputs, and then design a divide-and-conquer topology graph Transformer to reason lane-lane topology and lane-traffic topology. In the topology graph Transformer, the lane-lane part accepts lane features and processes them as a directed graph, while the lane-traffic part is treated as an undirected bipartite graph. After aggregating local information through message-passing graph neural networks (MPNNs) and global information via global attention layer in the two parts, the model employs MLPs to gather features of adjacent nodes and edge features to predict topology relationship categories. In this way, the model can learn the underlying guiding relationships between target objects, and thus obtain more accurate topology reasoning results. Moreover, we design a traffic scene-assisted reasoning module and incorporate it after the topology graph Transformer, allowing the model to constrain topology reasoning based on the practical significance of traffic objects.

To ensure the correct lane orientation, we further develop a point-wise matching strategy and integrate it into the lane detection module. Concretely, the point-wise matching strategy constructs ground truth for lane curve point sets with multiple equivalent arrangements. Through lane instance-level matching and curve point-wise matching constraints, the model decodes more accurate

lane instance embeddings, thereby improving its performance in lane detection and lane-lane topology reasoning.

The experimental results on two subsets of the benchmark dataset Openlane-V2 [44] demonstrate that TSTGT achieves state-of-the-art performance in driving scene topology reasoning task. Ablation studies are conducted to validate the effectiveness of our framework. The main contributions of this work can be summarized as follows:

- We propose an innovative topology reasoning framework, TSTGT, which can accurately detect target objects in multi-view images and derive precise topology relationships by employing divide-and-conquer graph information aggregation and imposing constraints on the orientation of lane targets. TSTGT achieves state-of-the-art performance on the benchmark Openlane-V2 dataset of this task.
- We devise a traffic scene-assisted topology graph Transformer to infer lane-lane topology and lane-traffic topology. It could effectively aggregate both local and global information of traffic objects and enhance the practical significance of lane-traffic topology, thus predicting high-quality topology relationships.
- We develop a point-wise matching strategy in the lane detector to constrain the orientation of detected lane objects. This encourages the detector to identify more accurate lane objects from visual features and improves the performance of lane-lane topology reasoning.

## 2 RELATED WORK

**Lane Topology Learning.** Lane Topology Learning has received abundant attention due to its pivotal role in autonomous driving. Earlier works used aerial images to generate a road graphs [1, 12, 18] or lane graphs [3, 17, 19, 52]. However, aerial images suffer from problems such as untimely updates and roads obscured by obstacles like trees, resulting in inaccurate results that do not match real-world driving conditions. As a result, it is becoming increasingly popular to use vehicle-mounted sensors to directly detect lane topology. STSU [6] proposes a DETR-like [9] neural network to detect centerlines and objects, and then derive them into a directed graph by a successive MLP module. On this basis, Can et al. [7, 8] introduce a minimal circle queries to provide additional supervision of the relationship, improving the estimation results of the lane graph. LaneGAP [29] designs a heuristic-based algorithm to learn from a set of lanes and build a lane topology. CenterLineDet [49] and TopoNet [26] use the road centerlines as vertices and design the graph model to update the topology. TopoMLP [48] designs a unified query-based framework for lane topology, which considers lane-traffic topology and lane-lane topology at the same time.

**HD Map Perception.** HD Map Perception is designed to understand the layout of driving scenarios, such as lanes, pedestrian crossings, and other traffic elements. In recent years, with the development of 2D-to-BEV methods [36], studies have focused on segmentation and vectorized methods. Chen et al. [11]; Zhou & Krähenbühl [50]; Li et al. [27]; Liu et al. [33] generate a rasterized map by performing BEV semantic segmentation. HDMapNet [25] uses heuristics and complex post-processing to group and vectorize the segmented map. VectorMapNet [32] serves as the

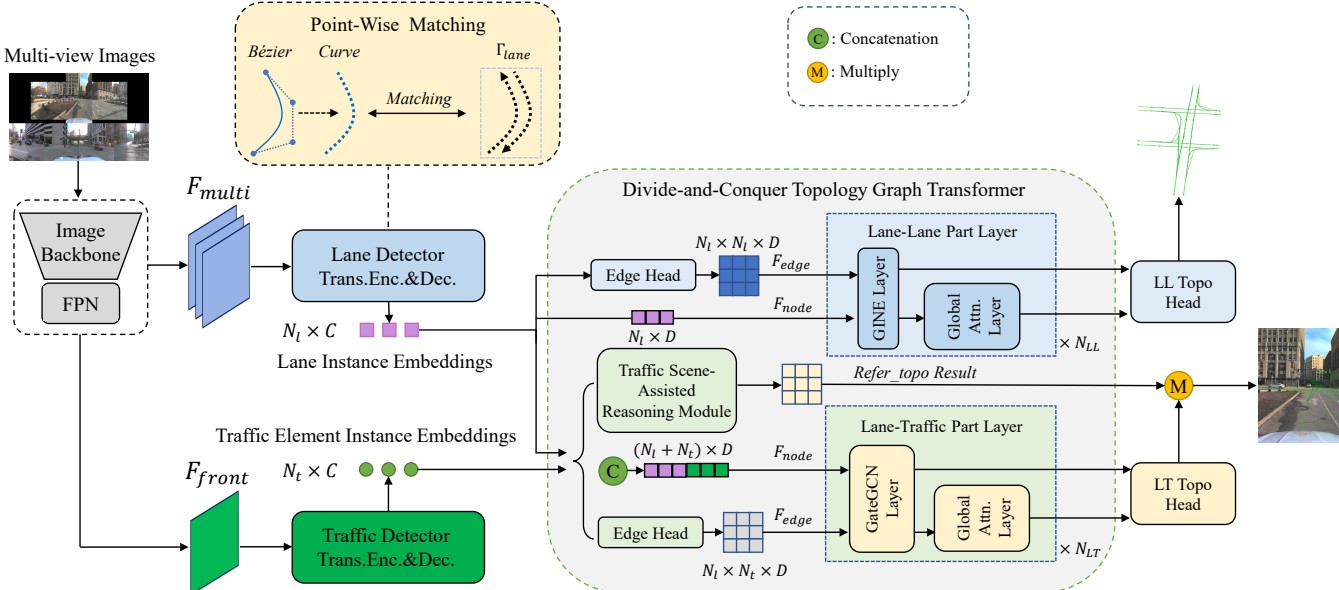

**Figure 2: The overview of the proposed TSTGT method. It mainly consists of four parts: the image backbone and FPN, the lane detector with point-wise matching, the traffic element detector, the divide-and-conquer topology graph Transformer with the traffic scene-assisted reasoning module. The point-wise matching strategy matches the lane curve points with all possible equivalent arrangements $\Gamma_{lane}$ of lane points. The divide-and-conquer topology graph Transformer receive corresponding instance embeddings and aggregate object information through corresponding MPNN (GINE, GatedGCN) layers and global attention layers. Simultaneously, the traffic scene-assisted reasoning module is utilized to fully leverage scene information to assist in topology modeling.**

first end-to-end framework, which adopts a two-stage framework and directly represents each map element as a sequence of points then utilizes auto-regressive decoder to predict them. MapTR [30] further explores a unified permutation-based modeling approach for the sequence of points to eliminate the modeling ambiguity. Since vectorization also enriches the direction information of the lanelines, the vectorization-based method can easily adapt to the perception of the centerline by alternate supervision. InstaGraM [42] builds map elements as a graph by predicting the vertices first and then using a GNN method to detect edges. TopoNet [26] leverage instance-level feature transmission with graph neural networks to extract map prediction hints.

**Graph Neural Network.** Graph Neural Networks, such as graph convolutional network (GCN) [23], GraphSAGE [15], GAT [43], and Graph Transformer (GPS) [40], are widely adopted to aggregate vertex features and extract information from graph data [41]. And they have contributed to remarkable achievements in various fields, e.g. recommender systems and video semantic segmentation. [10, 14, 37, 38]. Researchers in the autonomous driving community are also trying to utilize it to process unstructured data. Weng et al. [45, 46] introduce GNN to analyze the interaction between agent features for 3D multi-target tracking. LaneGCN [28] extracts the lane graph from the HD map. Others [13, 21, 22] model the relationship between moving agents and lanes as graphs to improve the trajectory prediction performance. TopoNet [26] uses GCN for driving scene understanding tasks, strengthens feature interaction, and introduces knowledge graphs to fuse semantic information.

Our method introduces a divide-and-conquer topology graph Transformer, which enhances the model's ability to explore potential relationships between objects by alternately aggregating local features and global features through MPNN and global attention layers, thereby leading to improved topology reasoning results.

## 3 METHOD

### 3.1 Overview

The overview of our proposed TSTGT is illustrated in Fig.2. It mainly consists of four parts: the image backbone and FPN, the lane detector with point-wise matching, the traffic element detector, the divide-and-conquer topology graph Transformer with the traffic scene-assisted reasoning module. During inference, given multi-view images from camera sensors, the image backbone first extracts the multi-view features $F \in R^{V \times C \times H \times W}$ and collects the front-view features $F^0 \in R^{1 \times C \times H \times W}$ among them, where $V$, $C$, $H$ and $W$ represent the view number, channel, height, and width of the features respectively. Then the multi-view features are fed into the lane detector and the front-view features are fed into the traffic detector. For the divide-and-conquer topology Transformer, the lane-lane part aggregates topology information in the lane instance embeddings output from the lane detector. The lane-traffic part is designed to understand the underlying control relationship between lane instance embeddings derived from the lane detector and the traffic element instance embeddings obtained from traffic detector. It leverages a traffic scene-assisted reasoning module to gather

pertinent scene data, aiding in the process of topology modeling. Finally, task-specific prediction heads are utilized separately for detecting precise target objects and reasoning accurate topology relationships.

## 3.2 Feature Extraction

For the surrounding images from multi-camera, we adopt a shared image backbone to extract the multi-level visual features of each image independently. Then we feed these multi-level features into the FPN to aggregate rich semantic information. Finally, the pyramid features are upsampled to the same size and stacked together as outputs.

## 3.3 Lane Detection

**Lane Detector.** We adopt a structure similar to PETR [31] as the lane decoder, which encodes 3D coordinates thus transforming multi-view features into 3D space, and incorporates 3D position embedding in order to enhance the perception of the query object in 3D space. Specifically, for the input multi-view features $F \in R^{V \times C \times H \times W}$, we first encode them using 3D position embedding to generate visual perceptual features $F^{3d}$, which are input as the key-value pairs into the Transformer. Then a set of learnable anchor points $Q^L \in R^{N_L \times 3}$ are randomly initialized, and after applying position encoding to them, they are projected into the feature space $R^{N_L \times C}$ by using a Linear layer in order to serve as the query embeddings for the PETRTransformer [31]. The lane query then interacts with the visual perceptual features $F^{3d}$ to generate a set of $N_L$ lane instance embeddings.

The previous approach used a set of Bézier curves to model lane. For the lane instance embedding from PETRTransformer, it chose to use two sets of MLPs to predict the lane categories and Bézier control points respectively, and convert the control points to lane points to make predictions about their position coordinates. However, lane objects are inherently sequential in character, and only using instance-level matching may make the direction of lane appear head-to-tail reversed, thus reducing the accuracy of topology prediction. Therefore, we design a point-wise matching strategy to improve the accuracy of lane matching.

**Point-Wise Matching.** Bézier curve is aimed to represent parametric curves by using an ordered set of control points $P_0$ through $P_n$:

$$B(t) = \sum_{i=0}^{n} P_i B_{i,n}(t), t \in [0, 1], \tag{1}$$

where $n$ represents the degree of the curve, and $B_{i,n}(t)$ is referred to as the Bernstein basis polynomial of degree n, as follows:

$$B_{i,n}(t) = \binom{n}{i} (1 - t)^{n-i} t^i, i = 0, \ldots, n. \tag{2}$$

Predicting the control points of Bézier curves with anchors of the Transformer can effectively abstract the curve features for better matching results [39]. However, the ground truth provided by instance-level matching consists of lane points with fixed arrangements, imposing restrictions on the model to learn the correct direction of the lanes. Inspired by MapTR [30], we re-interpolated and sampled the ground truth. For the given $N$ ground truth points, we use 3D linear interpolation sampling to obtain $N_p$ curve lane points,

and the two equivalent alignments of lane points were employed as the ground truth $A_p$ for point-wise matching. The equivalent alignments of the lane points are as follows:

$$\Gamma_{lane} = \{\gamma_0, \gamma_1\} = \begin{cases} \gamma_0(i) = i \% N_p, \\ \gamma_1(i) = (N_p - 1) - i \% N_p. \end{cases} \tag{3}$$

The predicted lane instance sequences $\hat{y}$ are assigned with ground truth lane instance sequences $y$ to locate the best prediction sequence as the positive sample $\hat{y}_{pos}$. After that, the lane control points predictions are converted into $N_p$ curved lane points $\hat{P}_{pos}$. Then, we perform point-wise matching on each lane instance assigned a positive label to find the optimal point match $\hat{\gamma}$ between the predicted curved lane points $\hat{P}_{pos(i)}$ and the ground truth point set $A_{p(i)}$. To constrain the matching results, the following point-wise matching cost is utilized:

$$\hat{\gamma} = \underset{\gamma \in \Gamma}{argmin} \sum_{j=0}^{N_p - 1} D_{Manhattan}(\hat{p}_j, p_{\gamma(j)}), \tag{4}$$

where $D_{Manhattan}(\hat{p}_j, p_{\gamma(j)})$ represents the Manhattan distance between the $j$-th point of the predicted lane curve points set $\hat{P}_{pos(i)}$ and the ground truth points set $A_{p(i)}$ with equivalent arrangement $\gamma$.

## 3.4 Traffic Elements Detection

Traffic elements are perceived from the front-view features, so this task can be considered as a 2D object detection task. Here we adopt the Deformable-DETR [51] as the traffic elements detector, which employs a set of randomly initialized reference points embeddings as the query $Q_t$ to interact with the front-view feature $F_0$ to generate a set of $N_t$ traffic element instance representations. Two MLPs are employed to them to predict the category and the bounding box of the traffic elements respectively.

## 3.5 Divide-and-Conquer Topology Graph Transformer

A natural idea for topology reasoning tasks is to model them using graph methods. Inspired by GPS [40], we design a divide-and-conquer topology graph Transformer (DCTGT) to model lane-lane and lane-traffic topology relationships respectively. The graph Transformer is stacked by several DCTGT layers, as shown in Fig.3. Each of them consists of a message-passing graph neural network (MPNN), a Transformer-like global attention layer and several MLPs. The MPNN, utilizing GINE [20] for the lane-lane part and GatedGCN [2] for the lane-traffic part, is employed to aggregate local neighborhood node information. Meanwhile, the global attention layers consolidate global node features, which allows for information propagation among all nodes in the graph ,thereby addressing to the extent issues such as over-smoothing and over-squashing of node information caused by the MPNN. Such the alternating learning approach between local and global information facilitates enhanced information perception capability for each node in the graph. The DCTGT layer could be formulated as follows:

$$\begin{aligned} F_{node}^{l+1} &= MLP^l(MPNN^l(F_{node}^l, F_{edge}^l) + GlobalAttn(F_{node}^l)), \\ F_{edge}^{l+1} &= MPNN^l(F_{node}^l, F_{edge}^l), \end{aligned} \tag{5}$$

Figure 3: The architecture of the divide-and-conquer topology graph Transformer layer.

where $F_{node}^l \in R^{N \times D}$, $F_{edge}^l \in R^{N \times N \times D}$ denote the $D$-dimensional node and edge features, respectively; $MPNN$ and $GlobalAttn$ refer to a MPNN and a global attention mechanism at layer $l$, each with its set of learnable parameters. $MLP^l$ stands for a 2-layer MLP block. The edge adjacency matrix $M_{adj} \in R^{N \times N}$ of the MPNN layer is defaluted to all ones to ensure that nodes fully utilize information exchange.

**Lane-Lane Part.** For the lane-lane topology reasoning, it can be considered as a problem of predicting edges categories from the known nodes features. We feed the lane instance embeddings $Q_l \in R^{N_l \times C}$ output from the lane detector into the lane-lane part of DCTGT (LLTGT) as the input node features $F_{node} \in R^{N_l \times D}$, and the initial edge sets are set to be the edge features $F_{edge} \in R^{N_l \times N_l \times D}$, which are aggregated from the neighbouring nodes features. After clustering the information from the $N_{ll}$ DCTGT layers, we feed the output node features $F'_{node} \in R^{N_l \times D}$ into the LL topology head composed of a 3-layer MLP to obtain predicted edge categories $C_{ll} \in R^{N_l \times N_l}$, which serve as the prediction results for the lane-lane topology relationship. The whole process can be represented by the following equation:

$$C_{ll} = MLP(LLTGT(F_{node}, F_{edge})). \tag{6}$$

**Lane-Traffic Part.** Due to the distinct semantic meanings represented by lane node features and traffic element node features, lane-traffic topology reasoning should be viewed as a problem of predicting edge categories in a bipartite graph based on known node features. Specifically, we concatenate the lane instance embeddings $Q_l \in R^{N_l \times C}$ and the traffic elements instance embeddings $Q_t \in R^{N_t \times C}$ and then feed them into the lane-traffic part of DCTGT (LTTGT) as the node features $F_{node} \in R^{(N_l+N_t) \times D}$. The edge sets between lane instances and traffic element instances are initialized as the edge features $F_{edge} \in R^{N_l \times N_t \times D}$, which are aggregated from adjacent node features. The edge adjacency matrix $M_{adj} \in R^{N_l \times N_t}$ are set to all-ones, allowing each lane-traffic element pair to have the opportunity to learn each other's information. Then, $F_{edge}$ and

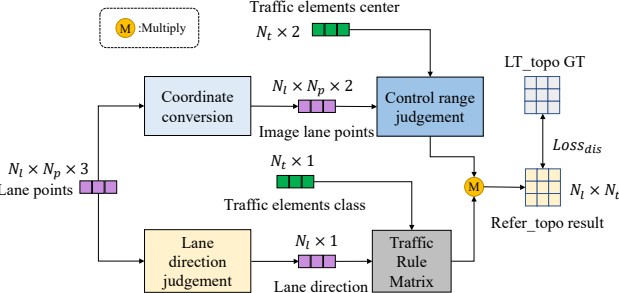

Figure 4: The structure of the traffic scene-assisted reasoning module.

$M_{adj}$ are expanded to the shape of $(N_l + N_t) \times (N_l + N_t)$ to fit the input of MPNN layer. Since the connection between lane nodes and traffic element nodes can be viewed as an undirected graph, meaning the topology relationship applies to both lane-traffic and traffic-lane pairs, $F_{edge}$ and $M_{adj}$ are symmetrically initialized for the lane-traffic element and traffic element-lane parts. Other parts are set to all-zeros to ensure the correct bipartite graph interaction. Due to the difference between lane instances and traffic element instances, we set them as two sets of node instances to ensure that they interact with global information in the global attention layers separately. After interacting with the information from the $N_{lt}$ LTTGT layers, we use the LT topology head consisting of two sets of 3-layer MLP to aggregate $F'_{node}$ and $F'_{edge}$ from the LTTGT, in order to obtain the predicted edge categories $C_{lt} \in R^{N_l \times N_t}$. The whole process can be represented by the following equation:

$$\begin{aligned} F'_{node}, F'_{edge} &= LTTGT(F_{node}, F_{edge}), \\ C_{lt} &= MLP(F'_{edge}, MLP(F'_{node})). \end{aligned} \tag{7}$$

**Traffic Scene-Assisted Reasoning Module.** Just reasoning the topology relationship based on data without incorporating the practical meaning will make the accuracy of topological inference decrease. In the previous method [48], only the lane-lane topology relationship is optimised, which uses L1 loss to constrain the positional coordinates of lane instances with topology associations. Nevertheless, the operation of optimising the lane-traffic topology relationship in conjunction with practical implications is missing. In this regard, we design a traffic scene-assisted reasoning module according to the construction way of Openlane-V2 [44] dataset and the traffic rules in the actual scenarios to overcome this difficulty, as shown in Fig.4.

Specifically, excluding the semantically unobservable category *unknown*, the traffic elements in the Openlane-V2 dataset can be classified into two groups, the traffic light class and the road sign class [44]. Among them, the traffic light class generates topology relationship for all lane instances within their control area, while the road sign class only takes effect on the lane instances in the corresponding direction within their control range. The judgement method for the control range of traffic elements and the direction of lane is outlined as follows.

Traffic element control range: We first transfer the lane points coordinates from the ego coordinate system to the front-view image coordinate system, and then calculate the minimum Manhattan distance between the centre coordinates of the traffic element bounding box $B_{tf}$ and the set of the lane curves points $P_{lane}$. The equations are as follows:

$$M_{dis} = min(D_{Manhattan}(B_{tf}, P_{lane})). \tag{8}$$

If $M_{dis}$ is in the $\sigma$ of the width of the image coordinate system, we consider that the lane instance is in the control range of the traffic element instance. Then we obtain the control range result $C_{con} \in R^{N_l \times N_t}$.

Direction of lane: The lane direction can be determined by the position coordinates of its start and end points. We obtain lane start point 3D coordinates $(x_0, y_0, z_0)$ and end point 3D coordinates $(x_n, y_n, z_n)$ from the lane detection result. For the vehicle ego coordinate system of Openlane-V2 dataset [44], the x-axis is positive forwards, the y-axis is positive to the left, and the z-axis is positive upwards. The lane direction is judged only in BEV, which means that the effect of the z-axis can be ignored. We use the threshold $\tau$ to delineate lane direction $D_{lane}$ as follows:

$$K_{lane} = \frac{y_n - y_0}{x_n - x_0},$$

$$D_{lane} = \begin{cases} left, & K_{lane} > \tau, \\ straight, & \|K_{lane}\| \leq \tau, \\ right, & K_{lane} < -\tau. \end{cases} \tag{9}$$

After obtaining the lane instance direction $D_{lane}$, we input traffic element category $C_{tf}$ and lane direction $D_{lane}$ into the traffic rule matrix $M_{tr}$ to obtain the traffic rule constraint topology result $C_{trc} \in R^{N_l \times N_t}$. The detailed introduction of $M_{tr}$ is in the Appendix. The traffic rule constraint topology result $C_{trc}$ is multiplied by the control range result $C_{con} \in R^{N_l \times N_t}$ to obtain the reference topology reasoning results for the traffic scene assistance $C_{ref} \in R^{N_l \times N_t}$. We compute the Manhattan distance between $C_{ref}$ and the ground truth of the lane-traffic topology relationship $G_{LT}$ to apply constraint, and multiply $C_{ref}$ by $C_{lt}$ to obtain the final lane-traffic topology reasoning result $C_{ltf} \in R^{N_l \times N_t}$.

## 3.6 Training Loss

Our model combines detection matching and topology reasoning matching for training, with the overall loss consisting of four components, *i.e.* lane detection loss, traffic detection loss, lane-lane topology reasoning loss and land-traffic topology reasoning loss:

$$\mathcal{L} = \mathcal{L}_{det_l} + \mathcal{L}_{det_t} + \mathcal{L}_{top_{ll}} + \mathcal{L}_{top_{lt}}. \tag{10}$$

**Lane detection loss** $\mathcal{L}_{det_l}$ is decomposed into a classification loss, a L1 loss and a point2point loss for lane point regression:

$$\mathcal{L}_{det_l} = \lambda_{cls}\mathcal{L}_{cls} + \lambda_{L1}\mathcal{L}_{L1} + \lambda_{pts}\mathcal{L}_{pts}, \tag{11}$$

where $\mathcal{L}_{cls}$ is a focal loss, and $\mathcal{L}_{pts}$ is a point2point loss of $N_l$ lane instance and $N_p$ curve points for each lane instance with positive class label ($C \neq \emptyset$), akin to MapTR [30]:

$$\mathcal{L}_{pts} = \sum_{i=0}^{N_l-1} \mathbf{1}_{\{C_i \neq \emptyset\}} \sum_{j=0}^{N_p-1} D_{Manhattan}(\hat{P}_{pos(i),j}, A_{p(i),\hat{y}_i(j)}). \tag{12}$$

**Traffic detection loss** $\mathcal{L}_{det_t}$ consists of a focal loss for classification, a L1 loss and a GIoU loss for bounding box regression:

$$\mathcal{L}_{det_t} = \lambda_{cls}\mathcal{L}_{cls} + \lambda_{L1}\mathcal{L}_{L1} + \lambda_{GIoU}\mathcal{L}_{GIoU}. \tag{13}$$

**Lane-lane topology reasoning loss** $\mathcal{L}_{top_{ll}}$ consists of a focal loss for binary classification, a L1 loss for constraining the coordinates of the start and end nodes of lanes with topology relationships:

$$\mathcal{L}_{top_{ll}} = \lambda_{cls}\mathcal{L}_{cls} + \lambda_{L1}\mathcal{L}_{L1}. \tag{14}$$

**Lane-traffic topology reasoning loss** $\mathcal{L}_{top_{lt}}$ consists of a focal loss for binary classification and a distance loss for traffic scene assistance topology reasoning:

$$\mathcal{L}_{top_{lt}} = \lambda_{cls}\mathcal{L}_{cls} + \lambda_{dis}\mathcal{L}_{dis}, \tag{15}$$

where the $\mathcal{L}_{top_{lt}}$ is defined as the Manhattan distance computed between traffic scene assistance topology result $C_{ref}$ and the ground truth of the lane-traffic topology relationship $G_{LT}$:

$$\mathcal{L}_{dis} = \sum_{i=0}^{N-1} D_{Manhattan}(C_{ref(i)}, G_{LT(i)}). \tag{16}$$

# 4 EXPERIENCES

## 4.1 Datasets and Metrics

**Datasets.** The experiments are conducted on the OpenLane-V2 [44], a comprehensive dataset designed for perception and reasoning tasks in autonomous driving scenes. OpenLane-V2 consists of two subsets, labeled as *subset_A* and *subset_B*, which are derived from Argoverse 2 [47] and nuScenes [4] datasets, respectively. Each subset contains 1,000 scenes annotated at a rate of 2Hz. It is noteworthy that *subset_A* encompasses seven views, whereas *subset_B* comprises six views.

**Evaluation Metrics.** The evaluation metrics of OpenLane-V2 dataset are divided into two parts: perception and reasoning. For the perception metrics, the DET score represents the standard mean average precision (mAP) used to evaluate the instance-level perception performance. In particular, $DET_l$ employs the Fréchet distance to quantify similarity, averaging over match thresholds set at {1.0, 2.0, 3.0}, while $DET_t$ calculates similarity using Intersection over Union (IoU) and averages across different traffic categories. Reasoning metrics also utilize an mAP metric known as the TOP score, tailored specifically for graph data. To encapsulate the combined impact of primary detection and topology reasoning, the OpenLane-V2 Score (OLS) is employed as follows:

$$OLS = \frac{1}{4}[DET_l + DET_t + f(TOP_{ll}) + f(TOP_{lt})], \tag{17}$$

where DET and TOP evaluate performance in perception and reasoning, respectively, while $f$ denotes the square root function.

## 4.2 Implementation Details

**Model Settings.** We employ different image backbones, including ResNet-50 [16], VOV [24], and Swin-B [34] for training and inference and realize a fair comparison with TopoMLP [48]. For feature extractor, each image undergoes resizing to a uniform resolution of 1550×2048, followed by downsampling at a ratio of 0.5, and the output channels number is set to $C = 256$. For lane detection, referencing TopoMLP [48], the specified region spans from -51.2m to 51.2m along the X-axis, from -25.6m to 25.6m along the Y-axis, and

**Table 1: Comparison with state-of-the-art methods on the Openlane-V2 *subset_A* dataset.**

| Method | Backbone | Epoch | $DET_l$ | $DET_t$ | $TOP_{ll}$ | $TOP_{lt}$ | OLS |
|---|---|---|---|---|---|---|---|
| STSU [6] | ResNet-50 | 24 | 12.7 | 43.0 | 0.5 | 15.1 | 25.4 |
| VectorMapNet [32] | ResNet-50 | 24 | 11.1 | 41.7 | 0.4 | 5.9 | 20.8 |
| MapTR [30] | ResNet-50 | 24 | 17.7 | 43.5 | 1.1 | 10.4 | 26.0 |
| TopoNet [26] | ResNet-50 | 24 | 28.5 | 48.1 | 4.1 | 20.8 | 35.6 |
| TopoMLP [48] | ResNet-50 | 24 | 28.3 | 50.0 | 7.2 | 22.8 | 38.2 |
| TSTGT | ResNet-50 | 24 | **29.0** | **50.5** | **12.1** | **23.5** | **40.7** |
| TopoMLP [48] | ResNet-50 | 48 | 29.6 | 50.4 | 9.6 | 24.1 | 40.0 |
| TSTGT | ResNet-50 | 48 | **31.2** | **51.4** | **14.9** | **25.0** | **42.8** |

**Table 2: Comparison with state-of-the-art methods on the Openlane-V2 *subset_B* dataset.**

| Method | Backbone | Epoch | $DET_l$ | $DET_t$ | $TOP_{ll}$ | $TOP_{lt}$ | OLS |
|---|---|---|---|---|---|---|---|
| STSU [6] | ResNet-50 | 24 | 8.2 | 43.9 | 0.0 | 9.4 | 21.2 |
| VectorMapNet [32] | ResNet-50 | 24 | 3.5 | 49.1 | 0.0 | 1.4 | 16.3 |
| MapTR [30] | ResNet-50 | 24 | 15.2 | 54.0 | 0.5 | 6.1 | 25.2 |
| TopoNet [26] | ResNet-50 | 24 | 24.3 | 55.0 | 2.5 | 14.2 | 33.2 |
| TopoMLP [48] | ResNet-50 | 24 | 26.6 | 58.3 | 7.6 | 17.8 | 38.7 |
| TSTGT | ResNet-50 | 24 | **27.5** | **60.5** | **13.7** | **18.9** | **42.1** |

from -8m to 4m along the Z-axis. The lane query number is set to $N_l = 300$, and the number of lane Bézier control points is configured as 4. Throughout the instance-level matching process, lane control points are transformed into 11 curve points for loss calculation. The configuration of PETRTransformer is based on PETR [31]. For traffic detection, all settings are identical to TopoMLP [48]. For topology reasoning, The lane-lane part employs 3 DCTGT layers, while the lane-traffic part employs 6. In the traffic scene-assisted reasoning module, the control range threshold $\sigma$ is set to 0.2, and the lane direction threshold $\tau$ is set to 0.1.

**Training Details.** All experiments are conducted on 8 NVIDIA A800 GPUs. Unless otherwise specified, the batch size of the model is set to 8, and the number of training epochs is set to 24 for fair comparison with TopoMLP [48]. The model is optimized using the AdamW optimizer [35] with a weight decay of 0.01. The initial learning rate of $4 \times 10^{-5}$ for the image backbone, and $2 \times 10^{-4}$ for the rest parts. Throughout the inference process, our model provides a maximum of 300 lane outputs for evaluation purposes. For lane detection loss $L_{det_l}$, the weights for each component loss are set as $\lambda_{cls} = 1.5$, $\lambda_{L1} = 0.02$, $\lambda_{pts} = 5 \times e^{-3}$. For traffic detection loss $L_{det_t}$, the weights allocated to each component of the loss are as follow: $\lambda_{cls} = 1.0$, $\lambda_{L1} = 2.5$, $\lambda_{GIoU} = 1.0$. For the lane-lane topology loss, the coefficient for the classification part is set as $\lambda_{cls} = 5$, and for the L1 loss is set as $\lambda_{L1} = 0.1$. The coefficients for the lane-traffic topology loss are set as $\lambda_{cls} = 5$, $\lambda_{dis} = 0.075$.

## 4.3 Comparison with State-of-the-Art Methods

**Openlane-V2 set.** We conduct a comprehensive comparison between our method, TSTGT, and several state-of-the-art approaches including STSU [6], VectorMapNet [32], MapTR [30], TopoNet [26],

**Table 3: Ablation study of different components of the proposed TSTGT on Openlane-V2 *subset_A* dataset. PWM denotes point-wise matching strategy, and DCTGT denotes divide-and-conquer topology graph Transformer.**

| Method | PWM | DCTGT | $DET_l$ | $DET_t$ | $TOP_{ll}$ | $TOP_{lt}$ | OLS |
|---|---|---|---|---|---|---|---|
| Baseline | | | 28.3 | 50.0 | 7.2 | 22.8 | 38.2 |
| TSTGT | ✓ | | 28.9 | 50.4 | 8.2 | 23.2 | 39.0 |
| TSTGT | | ✓ | 28.4 | 49.4 | 11.0 | 23.4 | 39.8 |
| TSTGT | ✓ | ✓ | **29.0** | **50.5** | **12.1** | **23.5** | **40.7** |

and TopoMLP [48]. The result of the comparison on *subset_A* are reported in Table 1. It can be observed that our approach, utilizing the ResNet-50 [16] backbone, outperforms other methods with an OLS score of 40.7. Notably, compared to TopoMLP, our method exhibits markedly superior topology reasoning accuracy, with 29.0 compared to 28.3 on $DET_l$ and 50.5 compared to 50.0 on $DET_t$. Additionally, it achieves respectable detection accuracy, with 12.1 compared to 7.2 on $TOP_{ll}$ and 23.5 compared to 22.8 on $TOP_{lt}$. The experimental results of employing VOV and Swin-B as backbone are provided in the Appendix.

In Table 2, the performance evaluation on OpenLane-V2 *subset_B* further reinforces our findings. Our innovative TSTGT outperforms its counterparts across all metrics while employing the ResNet-50 backbone. Notably, it demonstrates a considerable advantage over TopoMLP in terms of topology performance, with substantial differences evident: 27.5 compared to 26.6 on $DET_l$, 60.5 compared to 58.3 on $DET_t$, 13.7 compared to 7.6 on $TOP_{ll}$, and 18.9 compared to 17.8 on $TOP_{lt}$. Additionally, the integration of more potent backbones leads to further performance enhancements. Overall, these results validate the effectiveness of our proposed TSTGT for driving scene topology reasoning task and demonstrate that our method achieves state-of-the-art performance.

Fig.5 presents the visualization of TSTGT. The lane detection results are displayed in multi-view images, while the lane-lane topology reasoning results are shown in the BEV. The traffic detection results and the lane-traffic topology reasoning results are displayed in the front-view image. Despite the complexity of the traffic scenes, TSTGT can still accurately perform detection and topology reasoning.

## 4.4 Model analysis

In this part, we perform extensive ablation experiments to investigate the influence of the core components of our TSTGT as well as the impact of different model settings. All of the experiments are conducted on the Openlane-V2 *subset_A* dataset.

**Components Analysis.** To explore the influence of the key components of our model, we first build a baseline model similar to TopoMLP [48], which is the TSTGT without point-wise matching strategy and divide-and-conquer topology graph Transformer (DCTGT). As shown in Table 3, the baseline model obtains an OLS score of 38.2. When we add point-wise matching strategy on the baseline model, the special TSTGT achieves an OLS score of 39.0, which is 0.8 higher than the baseline. The improvement proves the effectiveness of the point-wise matching strategy. When only the

GT(left), Prediction(right)     GT(left), Prediction(right)

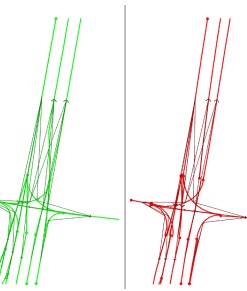
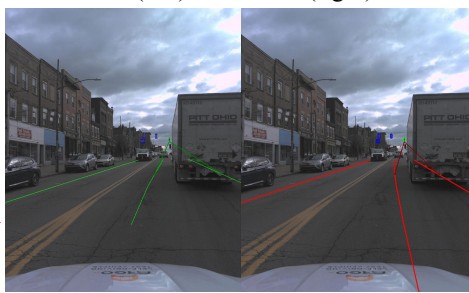

(a) lane detection     (b) lane-lane topology    (c) traffic detection and lane-traffic topology

Figure 5: Visualization of TSTGT. In (c), traffic elements of different categories are represented by bounding boxes of different colors, while lane-traffic topology relationships are indicated by red arrows.

Table 4: Model analysis of different settings in TSTGT.

| Method | Settings | $DET_l$ | $DET_t$ | $TOP_{ll}$ | $TOP_{lt}$ | OLS |
|--------|----------|---------|---------|------------|------------|-----|
| **Curve Points Number** | | | | | | |
| TSTGT | 11 | 28.2 | 50.3 | 11.4 | 21.7 | 39.7 |
| TSTGT | 30 | 29.0 | 50.5 | 12.1 | 23.5 | **40.7** |
| TSTGT | 50 | 26.1 | 50.4 | 11.8 | 23.9 | 40.1 |
| **Edge Features Initialization of DCTGT** | | | | | | |
| TSTGT | Aggregated | 29.0 | 50.5 | 12.1 | 23.5 | **40.7** |
| TSTGT | All-one | 28.7 | 49.2 | 11.7 | 23.4 | 40.1 |
| **Interaction Mode of LTTGT** | | | | | | |
| TSTGT | Heterogeneous | 29.0 | 50.5 | 12.1 | 23.5 | **40.7** |
| TSTGT | Homogeneous | 25.2 | 51.2 | 11.6 | 22.9 | 39.6 |

DCTGT is imposed on the baseline, the OLS score of the special TSTGT reaches to 39.8, which means the module brings a 1.6 gain and validates the superiority of the module. Equipped with both components, our TSTGT can achieve the best performance.

**Point-Wise Matching.** In this study, we explore the impact of different settings in the point-wise matching strategy. A critical setting involves the determination of the number of sampled points along the lane curve when converting Bézier control points to lane curve points. As shown in Table 4, there is a noticeable improvement in lane detection performance and other task performances when the number of sampled points increases from 11 to 30. However, it's worth noting that further increases do not enhance performance. This is because an excessive number of sampled points increases the burden on the model to match lane points while decreasing the abstraction level of the Bézier curve representation for the lane. Therefore, the number of sampled points along the lane curve is set to 30 to strike a balance between model efficiency and optimal performance.

**Divide-and-Conquer Topology Graph Transformer.** The impact of different settings of divide-and-conquer topology graph Transformer (DCTGT) on topology inference performance is also worth noting. First is the initialization method for edge features in the graph Transformer. We designed two initialization methods:

one, named "Aggregated", utilizes MLPs aggregation to aggregate node features adjacent to the edge as the initial edge features, while the other, named "All-one", adopts a similar approach to ARGNP [5], utilizing all-ones features as the initial edge features. The former setting is adopted in TSTGT and the later is a common paradigm. The experimental results are reported in Table 4. It can be seen that the TSTGT with the default setting performs better than the one with the later setting, which means that the "Aggregated" strategy enables better perception of node features, leading to superior topology reasoning performance.

The second setting is the interaction mode of the lane-traffic part of DCTGT (LTTGT). In our model, We perform global attention operations respectively on the lane features and traffic element features inputted into the LTTGT as *heterogeneous* features. Another approach involves treating them as *homogeneous* features and applying global attention operations uniformly. The results are presented in Table 4. It can be observed that employing the *heterogeneous* strategy for global attention helps to avoid confusion of information from different types of objects.

## 5 CONCLUSION

In this paper, we propose a novel framework for driving scene topology reasoning, termed TSTGT, to address the issue of insufficient utilization of object and scene information in the existing topology modeling process. Our approach involves designing a divide-and-conquer topology graph Transformer, which effectively aggregates local and global information of traffic objects, consequently obtaining more accurate topology reasoning results. Moreover, we devised a traffic scene-assisted reasoning module to enhance the practical significance of lane-traffic topology. For lane detection, we developed a point-wise matching strategy to ensure the accuracy of lane directions, further improving the performance of topology reasoning and detection. Experimental results on Openlane-V2 dataset validate the superiority of our TSTGT over state-of-the-art methods and the effectiveness of our proposed modules. We aspire for this work to inspire subsequent research endeavors.

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
