# OpenReview forum: "Driving Scene Understanding with Traffic Scene-Assisted Topology Graph Transformer"
_acmmm.org/ACMMM/2024/Conference — MM2024 Poster_

### Official Review · Reviewer_neD9 · 2024-05-19

**Rating:** 4
**Confidence:** 3

**Summary:**

This paper proposes a novel topology reasoning model called TSTGT. By employing divide-and-conquer graph information aggregation and imposing constraints on the orientation of lane targets, TSTGT achieves competitive performance on OpenLane-V2 benchmark. In detail, a traffic scene-assisted topology graph Transformer is devised to infer lane-lane topology and lane-traffic topology and a point-wise matching strategy is proposed to constrain the orientation of detected lane objects.

**Strengths:**

1. The main idea of the paper is clear, focusing on the incorporation of scene information into topology reasoning. Many design choices in the paper are also well-motivated.

2. The paper presents a well-organized related work and states how the contributions of this work contrasts w.r.t. relevant literature. The Fig. 1 is particularly helpful as it clearly illustrates how the proposed method differs from the existing work.

3. The paper includes abundant experiments to showcase the effectiveness of the proposed method. It reports the results on real-world datasets, including OpenLane-V2 subset_A and subset_B, all achieving competitive performance.

**Limitations:**

1. Even though TSTGT surpasses some classic models (TopoNet, TopoMLP), it is still far from SOTA performance. In Tab. 1, the OLS score of TSTGT on OpenLane-V2 subset_A is only 42.8 while the OLS scores of SOTA models [1] on the official benchmark [2] are over 55.

2. The performance improvements brought by the designed modules are too weak. In Tab. 3, both PWM and DCTGT only bring about a 1~2 improvement in OLS score, which is negligible and can be achieved by parameter tuning.

3. As an online perception task, reporting runtime is necessary.


[1] Wu D, Jia F, Chang J, et al. The 1st-place solution for cvpr 2023 openlane topology in autonomous driving challenge[J]. arXiv preprint arXiv:2306.09590, 2023.

[2] https://opendrivelab.com/challenge2023/#openlane_topology

**Suitability:**

2

---

### Official Review · Reviewer_syKy · 2024-05-25

**Rating:** 3
**Confidence:** 3

**Summary:**

This paper presents a solution for Driving Scene Understanding, which is in detail with a traffic Scene-Assisted Topology Graph Transformer. The performance on the  Openlane-V2 𝑠𝑢𝑏𝑠𝑒𝑡_𝐴 dataset has shown the improvements. But Tables 1 and 2 seem different from Table 3 in the reference [44]. This makes the paper less convincing.

**Strengths:**

1 Using the divide-and-conquer strategy to address lane-lane topology and lane-traffic topology seems interesting.

**Limitations:**

1 There are concerns concerning the results provided by the author, in Table 3 of [44],  and the results in Tables 1 and 2 of the paper are different, which makes the results less convincing. Could the author explain the reason why the difference in results exists?

[44] Huijie Wang, Tianyu Li, Yang Li, Li Chen, Chonghao Sima, Zhenbo Liu, Bangjun
Wang, Peijin Jia, Yuting Wang, Shengyin Jiang, et al . 2024. Openlane-v2: A
topology reasoning benchmark for unified 3d hd mapping. Advances in Neural
Information Processing Systems 36 (2024).

**Suitability:**

3

---

### Official Review · Reviewer_gyez · 2024-05-25

**Rating:** 3
**Confidence:** 3

**Summary:**

In the task of Driving Scene Topology Reasoning, the authors propose a complex graph reasoning module, the Divide-and-Conquer Topology Graph Transformer and scene-assisted reasoning module, to effectively utilize the associations between scene information and traffic objects to model topological relationships. Additionally, the authors introduce a point-wise matching method to enhance Lane Detection performance. These two innovations together improve Driving Scene Topology Reasoning and achieve state-of-the-art results, with promising experimental outcomes.

**Strengths:**

1.The paper is articulated with clarity and effectively outlines the technical workflow in a coherent and logical sequence.
2.The method shows notable performance improvement in the Lane-Lane topology metric (Top_ll), which is very promising. According to ablation study, the method achieves a significant improvement in the Lane-Lane topology metric (Top_ll) (11.0 compared to 7.2), indicating that the complex Lane-Lane topology module design indeed works.

**Limitations:**

1. As it mentioned that one of contributions is the inference of direction between lanes, yet the paper does not provide sufficient experimental results or visual demonstrations to directly showcase the effectiveness of this reasoning. I recommend incorporating visualizations that can clearly illustrate how these directional relationships are inferred and validated within the framework.
2. Lacks the computational efficiency of the proposed method to assess the applicability in real-world scenarios. Comparisons with existing methods on these aspects would also be helpful.
3. A detailed discussion comparing the performances of the two subsets is needed. The results show that the subset with more views performs better. What differences do the additional views introduce, and which module contributes most to this performance?
4. The improvement in Lane-Traffic Topology prediction is limited. Despite the introduction of a Traffic Scene-Assisted Reasoning Module and a Lane-Traffic Part Layer—key differences from TopoMLP[1] and primary innovations—these components do not significantly enhance Lane-Traffic Topology prediction performance. This limitation is evident from the ablation study, where the Top_lt metric increased only marginally from 22.8 to 23.4. The authors need to explain the reasons behind the discrepancy in the improvements observed in Top_ll and Top_lt.
5. According to TopoMLP[1] insights, the performance of Topology prediction is limited by the traffic detector performance, leading them to use YOLOv8 to generate proposals. Therefore, is the limited improvement in Lane-Traffic Topology prediction due to the detector? If the authors also use YOLOv8 and achieve remarkable improvement in Top_lt, then the designed methodology will be truly convincing.
6. Evaluation on the Enhanced Top metric is needed. TopoMLP[1] claims that the previous TOP score has a loophole where setting the confidence of unmatched instances to 1 can achieve remarkable improvements. The authors of TopoMLP [1] proposed an enhanced Top score, which has been accepted by the challenge organizers of OpenLane-V2. It seems that the initial Top score was used; it would be better to provide an evaluation using the enhanced Top score.
7. A discussion on hyper-parameters, such as the control range threshold, is needed. The authors introduce a threshold to limit the effect of road sign classes on lane instances only within a certain range. However, it is unclear why setting this threshold to 0.2 is considered reasonable for all types of road signs.
[1]Dongming Wu, Jiahao Chang, Fan Jia, Yingfei Liu, Tiancai Wang, and Jianbing Shen. 2023. TopoMLP: An Simple yet Strong Pipeline for Driving Topology Reasoning. arXiv preprint arXiv:2310.06753 (2023).

**Suitability:**

2

---

### Official Review · Reviewer_RzwB · 2024-05-26

**Rating:** 4
**Confidence:** 3

**Summary:**

This paper proposes a new topology reasoning framework TSTGT. It uses a divide-and- conquer topology graph Transformer to infer lane-lane and lane-traffic topology relationships separately. In addition, this paper also designed a traffic scene-assisted reasoning module and combined it with a topology graph Transformer. Meanwhile, in terms of lane detection, a point-wise matching strategy has been developed to infer lane centerlines with the correct direction.

**Strengths:**

The paper demonstrates that TSTGT achieves state-of-the-art performance on the Openlane-V2 benchmark, indicating the effectiveness of the proposed methods.
The paper introduces a novel topology reasoning framework, TSTGT, which combines a divide-and-conquer strategy with a graph transformer for improved traffic scene understanding.

**Limitations:**

In the Openlane-V2 subset_A dataset, when you set the epoch to 48, the metric is further improved. Can you inform us of the best achievable indicators as the epoch increases? Why did not an experiment with epoch=48 be conducted in Openlane-V2 subset_B?
Can you provide experiments using other backbones rather than just Resnet-50? And it would be better to explain why you chose Resnet-50 as your backbone? What’s more, providing the experiments of different backbones in paper’s body part rather than appendix may be better.
Visual comparisons with SOTA methods on different datasets should be given.
How are the hyperparameters in each loss selected? Including the weights for each component in    loss 𝐿𝑑𝑒𝑡𝑙 and  𝐿𝑑𝑒𝑡𝑡, and the coefficients for other two losses.

**Suitability:**

2

---

### Meta-Review · Area_Chair_3xAg · 2024-06-30

**Recommendation:** Accept (Poster)
**Confidence:** 5

**Metareview:**

The paper introduces TSTGT, focuses on driving scene understanding task. The authors propose a traffic scene-assisted topology graph Transformer with divide-and-conquer design for topology reasoning and a point-wise matching strategy for accurate lane detection. Experiments on OpenLane-V2 datasets prove the effectiveness of the proposed method.

The preliminary ratings are (ba br br ba). Key concerns are:
1. Backbone and hyperparameters settings.
2. Missing visual comparision.
3. Marginal improvement on TOP_lt and OLS by the proposed module.
4. Marginal improvement from previous SOTA TopoMLP.

After author rebuttal, the final ratings are (ba ba r ba). Reviewer RzwB and neD9 keep their scores and reviewer syKy raise his score as their concerns are resolved. Reviewer gyez seems to be against the acceptance of this paper as 1) missing visualization and 2) unready claim of generalization of the proposed method. In the view of AC, 1) is viable and could be expected given enough rebuttal time, and actually, the visual comparision between TopoMLP and proposed methods was submitted in the supplementary material and 2) is not standing as the paper does not claim generalization across the two datasets.

Based on the comments above, AC decides to accept the paper since the paper shows significant improvement thanks to the addressed concerns. The authors should improve the manuscripts according to the comments above.